# Neurological Manifestations in Pediatric COVID-19 Patients Hospitalized at King Abdulaziz University Hospital, Jeddah, Saudi Arabia: A Retrospective Study

**DOI:** 10.3390/children9121870

**Published:** 2022-11-30

**Authors:** Moustafa A. Hegazi, Fajr A. Saeedi, Ali F. Atwah, Mohamed H. Sayed, Asala A. Albeladi, Shahad B. Alyoubi, Razan A. Aljudibi, Shahad M. Alyaslami, Abdullah T. Alkathiry, Abobakr A. Abdelgalil

**Affiliations:** 1Department of Pediatrics, Faculty of Medicine in Rabigh, King Abdulaziz University, Jeddah 21589, Saudi Arabia; 2Department of Pediatrics, Mansoura University Children’s Hospital, Mansoura 35516, Egypt; 3Department of Pediatrics, Faculty of Medicine, Cairo University, Cairo 12613, Egypt; 4Faculty of Medicine in Rabigh, King Abdulaziz University, Jeddah 21589, Saudi Arabia; 5Department of Pediatrics, King Abdulaziz University Hospital, Jeddah 21589, Saudi Arabia

**Keywords:** neurological manifestations, clinical features, outcome, hospitalized pediatric COVID-19, Saudi Arabia

## Abstract

There are limited data about neurological manifestations in pediatric COVID-19 patients from all over the world, including Saudi Arabia. This study was performed to identify characteristics of pediatric COVID-19 cases with neurologic involvement hospitalized at King Abdulaziz University Hospital (KAUH), Saudi Arabia. This retrospective cross-sectional study included hospitalized patients aged 0–19 years with confirmed SARS-CoV-2 from April 2020 to February 2022. The required data were retrieved from patients’ medical records. Ninety-four cases were included. The median ages of the studied group, those with neurological manifestations, and those without neurologic manifestations, were 6.5, 11.0, and 5.0 years, respectively. Neurological manifestations occurred in 29 COVID-19 patients (30.9%) with headache and decreased consciousness being the most common recorded manifestations in 8.5% and 6.4% of patients, respectively. Specific neurological manifestations were rare, as only two infants developed encephalopathy with fatal outcome. Most patients with and without neurological manifestations survived. Neuroimaging abnormalities were detected in 8 cases with neurological manifestations. Neurological manifestations were common in 31% of hospitalized pediatric COVID-19 cases. However, most of the neurological manifestations were mild and nonspecific, with headache being the most common one. Specific neurological manifestations were rare; however, pediatric COVID-19 patients, particularly young infants, were at risk of developing severe encephalopathy with fatal outcome.

## 1. Introduction

Coronavirus disease 2019 (COVID-19) is a respiratory infectious disease caused by severe acute respiratory syndrome coronavirus 2 (SARS-CoV-2) [1]. Until September 2022, about 599,071,265 confirmed worldwide cases of COVID-19, including 6,467,023 deaths worldwide, were reported by the World Health Organization (WHO) [2]. From January 2020 to September 2022, Saudi Arabia reported 813,284 confirmed cases of COVID-19 to the WHO [3]. Although COVID-19 is more severe in the adult population, it can cause several morbidities and can be potentially fatal in pediatric patients if they develop multisystem inflammatory syndrome in children (MIS-C) [4].

SARS-CoV-2 infection causes less severe symptoms in children compared to adults, as about 95% of pediatric COVID-19 cases were asymptomatic or had mild to moderate manifestations, with fever and respiratory symptoms being the most common manifestations. Severe COVID-19 with admission to PICU was recorded in 2% of pediatric patients [5]. However, other than respiratory manifestations, a wide spectrum of neurological manifestations and complications were reported in COVID-19 adult patients, ranging from mild symptoms such as headache, myalgia, dizziness, hyposmia, hypogeusia, and altered consciousness to more severe manifestations such as acute encephalopathy or encephalitis, seizures, Guillain-Barré syndrome (GBS), acute ischemic or hemorrhagic stroke, and acute transverse myelitis [6,7,8,9,10,11].

However, the reporting of pediatric COVID-19-related neurological involvement has mostly been derived from case reports or small case series. The most frequently-reported symptoms in descending order are headache, altered mental status, seizures, muscular weakness, and meningism. The recognized severe neurological manifestations or complications, in a descending order, are acute cerebrovascular accidents (ischemic and hemorrhagic stroke), reversible splenial lesions, GBS, pseudotumor cerebri, autoimmune encephalitis, meningoencephalitis, acute demyelinating encephalomyelopathy (ADEM), cranial nerves impairment, transverse myelitis, and severe encephalopathy. Most of the reported neurological complications were associated with a favorable outcome; nevertheless, several cases had a poor or even fatal outcome [12]. Possible underlying pathogenic mechanisms could be attributed to several hypotheses such as direct viral invasion of CNS [13], molecular mimicry immune response following COVID-19 infection, or hyper-inflammatory state [14].

Despite several reports of neurological involvement in adult COVID-19 patients, there are limited data about neurological manifestations in pediatric COVID-19 patients. Moreover, most of these neurological manifestations are described from case reports or small cases series [12]. Therefore, this study was performed to identify the characteristics, risk factors and outcome of neurological involvement in pediatric COVID-19 cases hospitalized at KAUH, Jeddah, Saudi Arabia. This can help to anticipate high risk patients for neurologic involvement in COVID-19 disease as early as possible to put the appropriate plan to provide them with prompt intervention and care.

## 2. Material and Methods

### 2.1. Study Design and Setting

This is a retrospective cross-sectional study which was conducted at a tertiary university hospital. Pediatric patients ≤19 years with confirmed SARS-CoV-2 infection and who were hospitalized at KAUH between April 2020 and February 2022 were included in this study. COVID-19 positive patients with incomplete data in their medical records were excluded. The admission policy for pediatric COVID-19 required confirmed SARS-CoV-2 infection in symptomatic cases with clinical suspicion of COVID-19 or, in asymptomatic cases, a history of contact with SARS-CoV-2 positive cases and underlying comorbidity, which is considered to be a high-risk factor for severe COVID-19 disease. The place of admission either to the pediatric ward or the pediatric intensive care unit (PICU) was determined by the attending physicians according to the severity of the case. This study was approved by the research ethics committee of KAUH (reference number 66–22), approval date: 3 February 2022. 

### 2.2. COVID-19 Diagnosis

COVID-19 diagnosis was confirmed using nasopharyngeal swab real-time reverse transcription polymerase chain reaction according to the World Health Organization-established protocol [15]. All pediatric patients clinically suspected to have COVID-19 or had a history of contact with SARS-CoV-2 positive cases were tested.

### 2.3. Data Collection

Patient medical records were extracted from the hospital information system and were evaluated for all available data, including basic demographic characters, clinical symptoms and signs, detailed neurological manifestations, comorbidities, disease course, duration of hospital stay, laboratory and radiological findings, place of admission (pediatric ward versus PICU), management/treatment plan, complications, and outcome. Patients with new onset neurological manifestations or exacerbation of an old neurological problem, that occurred at the time of SARS-CoV-2 infection or shortly after, were defined as COVID-19 cases with neurological manifestations after consultation with a pediatric neurologist. 

### 2.4. Data Analysis 

Data were analyzed by The Statistical Package for Social Sciences (SPSS) software version 25 (IBM corporation, Armonk, NY, USA). Nominal or categorial data were presented as numbers and percentages. Numeric data were presented as means, ranges, medians and interquartile range (IQR). Chi square (χ2) and Fisher exact tests were used to compare between nominal variables. Student’s t test was used to compare between means, Mann–Whitney U test was used to compare between medians, and *p* < 0.05 was considered as statistically significant.

## 3. Results

Ninety-four pediatric patients with confirmed COVID-19, who were hospitalized at KAUH, from April 28, 2020 to February 28, 2022, were included in this study. The median age was 6.5 years, the interquartile range (IQR) was from 11 months to 16 years and range was from 5 days to 19 years. COVID-19 affected all pediatric age groups including neonates, infants, children, and adolescents; however, there were significantly more cases (*p* ≤ 0.0001) in children and adolescents (67, 71.3%) than in neonates and infants (27, 28.7%). There was no significant gender predominance and females represented 51.1% of the studied group. More than half (55.3%) of the patients had been in contact with COVID-19 patients, and about half (47.9%) of the patients had been in contact with COVID-19 family members. 

The main presenting complaint was fever (25.5%), followed by respiratory complaints (22.3%). The majority of the patients (89.4%) were admitted to the ward. Comorbidities were reported in 35.1% of patients, with hematological cases being the most frequent (9.6%). Only three patients (3.2%) died. The first mortality was a one-month-old girl without underlying comorbidity who presented mainly with fever and mild upper respiratory symptoms but developed neurologic complications on the fifth day of illness in the form of progressive encephalopathy presenting with frequent seizures and loss of consciousness. Her laboratory abnormalities showed anemia, lymphopenia, and high ferritin and D-dimers levels. Neuroimaging studies revealed marked hypodensity of the basal ganglia, altered signal of the splenium of corpus callosum, and diffuse loss of supra and infratentorial grey matter. The second mortality was a seven-month-old girl who was previously diagnosed with a neurodegenerative white matter disease which presented mainly with fever and cough but developed progressive deterioration of consciousness. Her laboratory abnormalities showed lymphopenia, high levels of ferritin, D-dimer, CRP, AST, ALT, and BUN, and a low serum albumin level. Neuroimaging studies revealed an altered signal of the splenium of corpus callosum in addition to the previous evidence of her neurodegenerative white matter disease. The third mortality was an 11-year-old boy without any underlying comorbidity, who presented mainly with fever, cough, and sore throat but developed progressive respiratory distress and MIS-C with multiorgan damage/failure (i.e., respiratory, hepatic, and renal) without any neurological manifestations and died four days after admission to PICU. His laboratory abnormalities showed leukocytosis, neutrophilia, lymphopenia, thrombocytopenia with high levels of ferritin, D-dimer, CRP, AST, ALT, total and direct serum bilirubin, and serum creatinine. His chest X-ray revealed bilateral pneumonic infiltrations. The demographic and clinical characteristics of pediatric COVID-19 cases are summarized in Table 1.

Neurological manifestations associated with COVID-19 were found in 29 patients (30.9%). Headache was the most common symptom (27.6%) followed by decreased consciousness (20.7%). Seizures were exhibited by 13.8 % of the studied group. Neurological symptoms are summarized in Figure 1.

Regarding the laboratory data, median white blood cell count (WBCs) was 7.9 × 10^3^/mm^3^, leukocytosis was detected in 17% of patients, and leukopenia was found in 8.5% of patients. Lymphopenia was evident among 34% of patients. Ferritin, C reactive protein (CRP), and D-dimer levels were high in 20.2%, 29.8%, and 36.2% of patients, respectively. Laboratory findings of the whole studied group are presented in Table 2. Details of additional demographic, clinical data, vital signs, and weight for age and sex of the studied group are provided as Appendix A.

Radiological findings of abnormal pulmonary infiltrations in chest x-rays were detected in 16 out of 94 cases (17%). Neuroimaging studies (brain MRI/CT) were done for all 29 cases with neurological manifestations, with abnormal findings in eight cases (27.6%). The most frequent neuroimaging abnormalities were hypodensity of corpus callosum and basal ganglia (3/8, 37.5%), diffuse loss of grey matter (2/8, 25%), and diffuse thinning of peri-ventricular white matter of neurodegenerative disease (1/8, 12.5%).

The management plan and treatment protocol were individualized according to the presentation and severity of each case and included analgesics/antipyretics, antibiotics (mostly a third generation cephalosporine and azithromycin), steroids (dexamethasone), anticoagulants (enoxaparin), anticonvulsants, intravenous fluids, blood and albumin transfusion, antishock measures, and organ support (i.e., intubation and mechanical ventilation for respiratory failure)

The median and IQR of age of the COVID-19 cases with and without neurological manifestations were 11 (0.58–17) years and 5 (1–15.5) years, respectively, with a non-significant difference between the two groups (*p* = 0.4). No significant statistical difference was found between pediatric COVID-19 cases with and without neurological manifestations regarding the demographic characteristics, including age and gender, course of the disease, duration of illness, PICU admission, and length of hospital stay. Symptoms at initial presentation were present in 100% of patients with neurological manifestations and in 92.3% of those without neurological manifestations. No statistically significant difference was detected in preexisting comorbidities between the patients with and without neurological involvement (*p* = 0.58). Although it was not statistically significant, disease outcome was worse in patients with neurological involvement, where two out of the three deaths exhibited neurological manifestations during the COVID-19 disease course. Comparisons of clinical data between COVID-19 cases with and without neurological manifestations are presented in Table 3. More details about duration from the appearance of symptoms until admission, duration from the appearance of symptoms until COVID-19 confirmed diagnosis, and length of hospital stay and comparison of vital signs and weight for age and sex between pediatric COVID-16 cases with and without neurologic manifestations are provided as Appendix A. Comparison of categorial variables of laboratory and radiological findings between pediatric COVID-19 cases with and without neurological manifestations are provided as Appendix A.

Regarding the laboratory data, cases with neurological manifestations had a significantly higher red blood cell count, hemoglobin level, and prolonged activated partial thromboplastin time (aPTT) than the cases without neurological manifestations (*p* = 0.02, *p* = 0.049, and *p* = 0.03, respectively). There were only five cases with abnormally prolonged aPTT. Three cases developed neurological manifestations with abnormal neuroimaging, including the one-month-old case with progressive encephalopathy that experienced mortality. The other two cases had severe COVID-19 disease without neurological manifestations/complications.

Among the eight patients with neurological manifestations and abnormal neuroimaging, two (25%) had high CRP, three (37.5%) had prolonged aPTT, five (62.5%) had a high ferritin level, and seven (87.5%) had a high D-dimer level. Comparison of laboratory findings between pediatric COVID-19 cases with and without neurological manifestations are presented in Table 4.

## 4. Discussion

Knowledge about the wide spectrum of clinical manifestations and complications of SARS-CoV-2 infection in the pediatric age must be constantly updated to help deal with the COVID-19 pandemic. SARS-CoV-2 mostly causes respiratory illness; nevertheless, patients with SARS-CoV-2 infection have also experienced a broad range of neurological manifestations and consequences, either during the disease or after recovery [12].

Pediatric COVID-19 represented about 2–5% of total COVID-19 cases. Children seem to be less severely affected than adults, and are mainly affected with pulmonary symptoms [17,18,19]. Preexisting co-morbidity such as immunosuppression was found in children with severe COVID-19 [20,21].

Thus far, pediatric COVID-19-related neurological manifestations have been relatively rare, yet reports of neurological manifestations in this age group are increasing. Also, most of the knowledge on COVID-19-associated neurological manifestations is derived from small cases series, case reports, and systematic reviews of the literature to gather data from small pediatric COVID-19 studies describing neurological manifestations to have a considerable number of cases [12]. Therefore, this study was performed to identify the characteristics, risk factors, and outcome of pediatric COVID-19 cases with neurological involvement admitted to KAUH, Jeddah, Saudi Arabia.

In this study, the most common clinical manifestations at presentation were fever and respiratory symptoms. Expectedly, more than half of our patients had contact with positive COVID-19 patients; especially family members, which is consistent with an early COVID-19 study in Saudi Arabia [14]. Moreover, all patients with neurologic involvement were symptomatic at presentation. Only one 11-year-old boy (about 1%) had MIS-C with a fatal outcome, which is significantly lower than the prevalence of MIS-C of 36% which was found in other studies [21].

Totals of 17% and 34% of the studied group had leukocytosis and lymphopenia, respectively, but with no statistically significant difference between COVID-19 patients with and without neurological manifestation. A large study performed at 61 hospitals, which included 1695 patients, demonstrated that lymphopenia is a common finding in COVID-19 pediatric patients, especially those with life threatening neurologic involvement [22].

In this study, COVID-19-associated neurological manifestations were reported in 30.9% of the patients, which is consistent with the reported prevalence of neurological manifestations in about one third of pediatric COVID-19 patients [23]. Other studies reported lower prevalences (3.8% and 22%) of neurologic involvement in children from different ethnic groups with COVID-19 in the UK and USA, respectively [22,24]. Moreover, a systematic review of 3707 pediatric COVID-19 cases found that 15.6% of cases had nonspecific neurological manifestations, while only 1.0% of cases had specific neurological manifestations [25]. On the other hand, a higher prevalence was also reported, where 43% of hospitalized children with confirmed COVID-19 exhibited neurological manifestations [26]. Furthermore, the prevalence of neurological manifestations in children with MIS-C was variable and ranged from 11.3% in patients with any neurologic involvement to 58% in patients with nonspecific neurologic manifestations such as headache, irritability, and lethargy [27,28]. In comparison to the adult population with COVID-19, early reports from Wuhan revealed that 36.4% of hospitalized patients exhibited neurological manifestations associated with COVID-19 [7].

In this study, the median age of pediatric COVID-19 cases with neurological manifestation was 11 years, which is very similar to the median age of pediatric COVID-19 cases with associated neurological manifestations recorded in other studies [12,22].

Most of the neurologic manifestations detected in this study were mild and nonspecific, with headache being the most common symptom, and which is in agreement with several previous studies [12,22,26,29]. Furthermore, although headache associated with COVID-19 was more prevalent in children with MIS-C [24], none of our patients developed MIS-C except one 11-year-old boy who developed progressive respiratory distress and MIS-C with multiorgan damage but without neurological manifestation. In adults, headache and myalgia were the most common nonspecific neurological manifestations, and cerebrovascular events were the most frequent specific neurological complications associated with COVID-19 [29]. In this study, decreased or altered consciousness was found in 6 out of 94 (6.4%) pediatric COVID-19 patients, which was significantly lower than the rate detected in other studies that ranged from 32% to 42% [12,22], and which may be explained by more frequent mild nonspecific neurological manifestations among our pediatric COVID-19 patients compared to other studies which recorded more severe and specific neurological manifestations.

Many studies reported seizures over the course of pediatric COVID-19 and MIS-C with variable prevalence, where seizures developed in 23% of cases with neurological manifestations [24]. In this study, seizures occurred in 4 out of 94 (4.3%) pediatric COVID-19 patients which was lower than the detected rate of 16% reported in another study [22]. Similarly, this may be explained by more frequent mild nonspecific neurological manifestations/complications among our pediatric COVID-19 patients. Unlike adults, anosmia is not a common finding at the pediatric age. Anosmia was found in 3.2% of our patients, which is lower than the prevalence of anosmia of 10% in children under 5 years and 38% in children from 13–20 years that was found in one study [22]. However, a systematic review of the literature, mainly studies on adults, detected smell impairment in 11 out of 214 patients (5.1%) [29]. Other less frequent neurological manifestations such as irritability, muscular weakness, hallucination, and photophobia were also found in our study, which is consistent with previous pediatric studies [12,22].

Accordingly, headache, altered consciousness, and seizures were the most frequent manifestations detected in our pediatric COVID-19 patients with neurologic involvement, which is closely similar to the findings in other studies [12,22].

COVID-19-related severe neurological involvement such as encephalopathy, stroke, meningoencephalitis, and acute disseminated encephalomyelitis have been reported mainly in adults [7,30]. Severe neurological manifestations or complications previously recorded in pediatric COVID-19 infection, such as cerebrovascular accidents, ischemic stroke, intracerebral hemorrhage, GBS, pseudotumor cerebri, autoimmune encephalitis, meningoencephalitis, ADEM, cranial nerves impairment, and transverse myelitis [11], were not reported in this study. Fortunately, in this study, COVID-19-related severe neurologic manifestations/complications were only detected in two cases; a one month-old girl without preexisting comorbidity but who developed progressive encephalopathy and frequent seizures with loss of consciousness and fatal outcome, and a seven-month-old previously known girl with an underlying neurologic comorbidity of neurodegenerative white matter disease who presented mainly with fever and cough but developed progressive deterioration of her consciousness, also with a fatal outcome.

In this study, no significant differences were found in demographic, PICU admission, or length of hospital stay between COVID-19 patients with and without neurological manifestations. This can be mostly related to the mild and nonspecific neurological manifestations detected in pediatric COVID-19 cases with neurological involvement which does not reflect or correlate with a severe COVID-19 disease. 

No significant difference in preexisting comorbid conditions was found between patients with and without neurological manifestations (*p* = 0.58). Similarly, LaRovere et al. reported no difference in underlying comorbidities between COVID-19 patients with and without neurological manifestations [22]. A systematic review by Siracusa et al. revealed that a minority of COVID-19 patients with neurologic involvement had underlying comorbidities [12]. Furthermore, children with COVID-19-related neurological manifestations exhibited fewer underlying comorbidities than adults in another study [25].

Although there was no statistically significant difference of outcome between cases with and without neurological involvement, worse disease outcome occurred in cases with neurologic manifestations/complications, where two out of the three deaths had exhibited severe specific neurological manifestations during the COVID-19 disease course, which is consistent with previous studies [22,31]. Additionally, it is worth emphasizing an important point that, although severe specific neurological manifestations (encephalopathy and seizures) were rare and encountered in only two cases (one-month- and seven-month-old infants—one of them had a prior neurodegenerative white matter disease),—both cases ultimately developed a fatal outcome, which may predict a lethal effect of severe specific neurological manifestations/complications in pediatric COVID-19, particularly in young infants or those with preexisting severe neurologic illness.

In the current study, patients with neurological manifestations had significantly more prolonged aPTT than patients without neurological manifestations (*p*= 0.03). Prolonged PTT has been reported as an indicator of poor outcome in adult patients hospitalized with COVID-19 [32]. Similarly, two patients had prolonged aPTT in an Italian study that reported neurological manifestations in 32 out of 237 children suffering from COVID-19 [33]. Furthermore, an autopsy study of the spectrum of severe COVID-19 in children found prolonged aPTT in two of five autopsy cases. Interestingly, the mentioned study showed, for the first time, the presence of SARS-CoV-2 in the brain tissue of a child with MIS-C and acute encephalopathy [34]. Additionally, patients with neurological manifestations had a significantly higher RBC count and hemoglobin level than patients without neurological manifestations (*p*= 0.02 and *p* = 0.049, respectively), which is in agreement with the findings in another study on the neurological symptoms and signs associated with pediatric COVID-19 [35]. However, further studies are needed to confirm, explain, and correlate these findings with pediatric COVID-19-related neurological manifestations.

In this study, the eight cases who had neurological manifestations and abnormal neuroimaging were associated with prolonged PTT, high ferritin, and high D-dimer in 37.5%, 62.5%, and 87.5% of cases, respectively. These laboratory derangements may predict more severe neurologic involvement in pediatric COVID-19 cases. In the current study, the most frequently-detected neuroimaging abnormality was hypodensity of corpus callosum and basal ganglia, which is similar to the findings of other studies [12].

Although this is a single-center experience, this study has its strengths. First, it described the characteristics, risk factors, and outcomes of pediatric COVID-19-related neurological manifestations in a considerable number of cases, which is more than the collectively retrieved small number of cases from case reports and case series in previous studies. Second, this study included the different pediatric age groups (from neonates to adolescents). Finally, this study showed pediatric COVID-19-related neurological manifestations from Saudi Arabia, which is in an area of the world other than China, USA, and Europe, where most of the previous relevant studies were conducted.

However, this study has the limitations of being of retrospective nature; being a single-center study with the involvement of only hospitalized patients and a lack of long term follow up of the studied group. Moreover, some neurologic symptoms (e.g., anosmia) may be underreported by young children.

## 5. Conclusions

Neurological manifestations were fairly common in about one third of pediatric COVID-19 cases hospitalized at KAUH, Jeddah, Saudi Arabia. However, most of the neurological manifestations were mild and nonspecific, with headache being the most common. Severe specific neurological manifestations were rare; however, pediatric COVID-19 patients are at risk of developing encephalopathy and seizures even without severe infection or MIS-C, which were encountered in only two cases (one-month- and seven-month-old infants); one of them had a prior neurodegenerative white matter disease and both cases ultimately had fatal outcome, which may predict a risk of fatal outcome of severe specific neurological manifestations/complications in pediatric COVID-19 cases, particularly in young infants or those with preexisting severe neurologic illness. Prolonged aPTT, high ferritin and D-dimer levels, and abnormal neuroimaging were associated with severe neurological manifestations during the course of COVID-19. Further larger studies are still needed to confirm the findings of this study and to evaluate the long-term neurological sequelae of COVID-19 in the pediatric population.

## Figures and Tables

**Figure 1 children-09-01870-f001:**
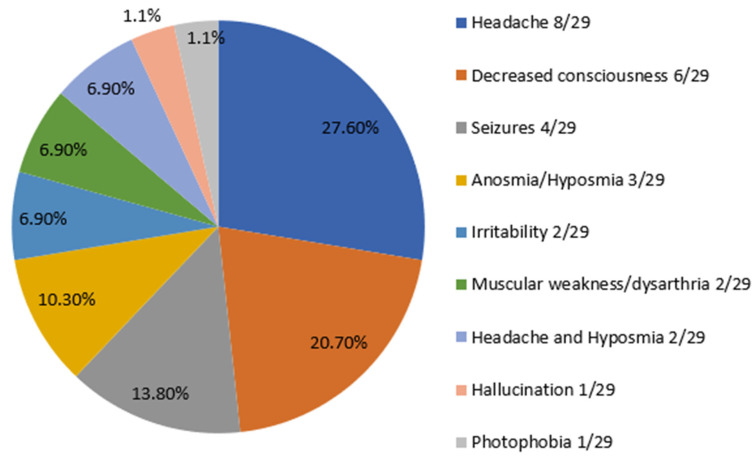
Frequency distribution of neurological symptoms in 29 pediatric COVID-19 patients with neurologic involvement.

**Table 1 children-09-01870-t001:** Demographic and clinical characteristics of the whole studied group (N = 94).

Variables	Number	%
**Gender**, females	48	51.1%
**Main presenting complaint (symptoms at initial presentation)** Fever Respiratory symptoms Fever and Respiratory symptoms Gastrointestinal symptoms Fever and neurological symptoms Fever and Gastrointestinal symptoms	24 21 19 10 8 7	25.5% 22.3% 20.2% 10.6% 8.5% 7.4%
**History of exposure to positive COVID-19**	52	55.3%
**Family member with positive COVID-19**	45	47.9%
**Place of admission** Ward PICU	84 10	89.4% 10.6%
**Comorbidities** Hematologic Cardiac Renal Undernutrition Leukemia Diabetes mellitus Chronic diarrhea Bronchial asthma Cystic fibrosis Immunodeficiency Epilepsy Cerebral palsy GBS Neurodegenerative disease Nephrotic syndrome	33 9 4 4 2 2 2 2 1 1 1 1 1 1 1 1	35.1% 9.6% 4.3% 4.3% 2.1% 2.1% 2.1% 2.1% 1.1% 1.1% 1.1% 1.1% 1.1% 1.1% 1.1% 1.1%
**Outcome** Survived Died	91 3	96.8% 3.2%

**Table 2 children-09-01870-t002:** Laboratory findings of the whole studied group (N = 94).

Variables	Measured Values	Variables by Category *	
			N%
**Total WBCs (×10^3^/mm^3^)**	(Median, IQR) 7.9 (5.4–12.3)	Normal WBC count Leukopenia Leukocytosis	70→8 16→74.5 8.5→17.0
**Neutrophils** **(×10^3^/mm^3^)**	(Median, IQR) 3.9 (2.3–7.4)	Normal neutrophil count Neutropenia Neutrophilia	77→81.9 12→12.8 5→5.3
**Lymphocytes (×10^3^/mm^3^)**	(Median, IQR) 2.4 (1.4–4.3)	Normal lymphocyte count Lymphopenia	62→66.0 32→34.0
**RBCs (×10^6^/mm^3^)**	(Mean ± SD, Range) 4.2 ± 0.97 (1,250,000–6,550,000)	Normal RBC count Anemia Polycythemia	56→59.6 34→36.2 4→4.3
**Hemoglobin (g/dL)**	(Mean ± SD, Range) 11.27 ± 2.59 (3.40–18.50)	Normal hemoglobin Low hemoglobin (anemia) High hemoglobin (polycythemia)	51→54.3 38→40.4 5→5.3
**Platelet count (×10^3^/mm^3^)**	(Median, IQR) 287 (211–382)	Normal platelet count Thrombocytopenia Thrombocytosis	81→86.2 7→7.4 6→6.4
**CRP (mg/dL)**	(Median, IQR) 3.74 (3.0–9.8)	Normal High	66→70.2 28→29.8
**Ferritin (ng/mL)**	(Median, IQR) 100 (74.5–123.8)	Normal High	75→79.8 19→20.2
**D-dimer (mg/L)**	(Median, IQR) 0.34 (0.24–0.66)	Normal High	61→63.8 33→36.2
**INR**	(Mean ± SD, Range) 1.10 ± 0.22 (0.90–2.38)	Normal High	84→89.4 10→10.6
**APTT Seconds**	(Mean ± SD, Range) 37.38 ± 7.65 (23.0–60.0)	Normal High	89→94.7 5→5.3
**ALT (U/L)**	(Median, IQR) 32 (25–45)	Normal High	78→83.0 16→17.0
**AST (U/L)**	(Median, IQR) 8 (22–34)	Normal High	65→69.1 29→30.9
**Total serum bilirubin (umol/L)**	(Median, IQR) 7.0 (5.0–11.0)	Normal High	82→87.2 12→12.8
**Serum Albumin (gm/L)**	(Mean ± SD, Range) 38.3 (4–55.4)	Normal High	82→87.2 12→12.8
**BUN (mmol/L)**	(Median, IQR) 3.5 (2.8–4.5)	Normal High	84→89.4 10→10.6
**Serum creatinine (µmol/L)**	(Median, IQR) 34 (22–50.3)	Normal High	80→85.1 14→14.9

* Reference values/ranges for WBCs, neutrophils, lymphocytes, and other lab tests were considered in relation to age and sex (whenever applicable) from Lo, S.F. 2016 [16].

**Table 3 children-09-01870-t003:** Comparison of clinical data between pediatric COVID-19 cases with and without neurological manifestations.

Variables	COVID-19 Cases without Neurologic Manifestations N = 65 N%	COVID-19 Cases with Neurologic Manifestations N = 29 N%	Significance *p*-Value
**Gender** *Male* *Female*	33→50.8 32→49.2	13→44.8 16→55.2	0.60 ^a^
**Nationality** *Saudi* *Non-Saudi*	31→47.7 34→52.3	10→34.5 19→65.5	0.23 ^a^
**Main initial presentation** Asymptomatic Symptoms at initial presentation Fever Respiratory symptoms Fever and Respiratory symptoms Gastrointestinal symptoms Fever and neurological symptoms Fever and Gastrointestinal symptoms	5→7.7 60→92.3 18→27.7 15→23.1 14→21.4 8→12.3 0→0 5→7.7	0→0 29→100 6→20.7 6→20.7 5→17.2 2→6.9 8→27.6 2→6.9	NA
**History of exposure to SARS-CoV2** *Yes* *No*	27→41.5 38→58.5	15→51.7 14→48.3	0.36 ^a^
**Family member with positive SARS-CoV2 infection** *Yes* *No*	33→50.8 32→49.2	16→55.2 13→44.8	0.96 ^a^
**Comorbidities (risk factors for severe COVID-19)** *Yes* *No*	24→36.9 41→63.1	9→31.0 20→69.0	0.58 ^a^
**Place of admission** *Inpatient ward* *PICU*	60→92.3 5→7.7	24→82.8 5→17.2	0.17 ^a^
**Outcome** *Survived* *Died*	64→98.5 1→1.5	27→93.1 2→6.9	0.17 ^a^

^a^ Chi-square test, NA: not applicable because Chi square cannot be applied when some cells have 0 count.

**Table 4 children-09-01870-t004:** Comparison of laboratory findings between pediatric COVID-19 cases with and without neurological manifestations.

Variables (Measured Value)	COVID-19 Cases without Neurologic Manifestations N = 65	COVID-19 Cases with Neurologic Manifestations N = 29	Significance *p*-Value
Red blood cells (Mean ±SD), range (×10^6^/mm^3^)	3.99 ± 1.0 (1.25–5.74)	4.5 ± 0.80 (2.56–6.55)	0.02 * ^a^
Hemoglobin (HB) (Mean ± SD), (gm/dL)	10.92 ± 2.71 (3.4–18.5)	12.06 ± 2.14 (7.8–18.5)	0.049 * ^a^
White blood cells (Median, IQR) (×10^3^/mm^3^)	7.96 (5.57–12.50)	7.80 (5.12–11.45)	0.44 ^b^
Neutrophils (Median, IQR) (×10^3^/mm^3^)	4.27 (2.28–7.46)	3.29 (2.45–6.80)	0.45 ^b^
Lymphocytes (Median, IQR) (×10^3^/mm^3^)	2.58 (1.42–4.66)	2.35 (1.21–3.87)	0.48 ^b^
Platelet count (Median, IQR) (×10^3^/mm^3^)	271 (206–387)	299 (210–376)	0.72 ^b^
CRP (Median, IQR) (mg/dL)	4 (3–8.28)	3.47 (3–21)	0.83 ^b^
Ferritin (Median, IQR) (ng/mL)	98 (71.5–120)	100 (81–154.75)	0.39 ^b^
D-dimer (Median, IQR) ug/mL	0.34 (0.27–0.63)	0.41 (0.20–0.73)	0.90 ^b^
INR (Mean ± SD)	1.10 ± 0.25 (0.90–2.38)	1.07 ± 0.13 (0.90–1.40)	0.97 ^a^
APTT (Mean ± SD) Seconds	36.22 ± 7.27 (23–60)	40.0 ± 7.95 (29.0–58.0)	0.03 * ^a^
Alanine aminotransferase (Median, IQR) (U/L)	28 (24–34)	29 (18–53)	0.73 ^b^
Aspartate aminotransferase (Median, IQR) (U/L)	33 (25–45)	31 (22–51)	0.46 ^b^
Total serum bilirubin (Median, IQR) (mg/dL)	7 (5–12)	6 (4–7.5)	0.02 * ^b^
Serum albumin (Median, IQR) (gm/L)	38.82 ± 7.41 (4–55.4)	37.0 ± 6.2 (19.10–47.3)	0.26 ^b^
BUN (Median, IQR) (mmol/L)	3.40 (2.8–4.55)	3.6 (2.7–4.35)	0.74 ^b^
Serum creatinine (Median, IQR) (umol/L)	34 (23–48)	38 (22–57)	0.49 ^b^

^a^ Independent *t*-test, ^b^ Mann–Whitney U test. * Significance < 0.05.

## Data Availability

Data is contained within the article and Appendix A.

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
