# Peer review of "Neurological Manifestations in Pediatric COVID-19 Patients Hospitalized at King Abdulaziz University Hospital, Jeddah, Saudi Arabia: A Retrospective Study"

_children, 2022, doi:10.3390/children9121870_

Round 1

Reviewer 1 Report

The paper lacks clear study design description, inclusion criteria, symptoms definitions.

 Authors of the paper and affiliation in missing in the main manuscript file

Table 1

Main presenting complaints:

Fever

Fever and gastrointestinal symptoms

gastrointestinal symptoms

These symptoms somehow overlap but I cannot track how

How 10 children with gastrointestinal symptoms be only 0.6% ?

Figure 1

Pi chart suggest that neurological symptoms sum to 100% but they are not. Does neurological symptoms overlap with one another?

Table 2

Why authors supply all the lab findings in the whole group when paper is about neurological manifestation.

The authors do not further discuss this data at all.

Table 3

Second row

COVID-19 cases with neurologic manifestations - FEVER 6 children

If these are children with neurologic manifestation and do have fever why they are not in the fever and neurological symptoms group.

According to table 1 there were only 8 children presented with neurological symptoms and finally 29. Did these symptoms occur later in the course of disease?

 Again I think the group definition is not clear

Author Response

Reply to Reviewer 1 Comments:

  1. The paper lacks clear study design description, inclusion criteria, symptoms definitions.

Response/Reply:

The reviewer didn’t give much details or precise examples of the lack of clarity in study design, inclusion criteria and symptom definition. However, the study design and inclusion criteria were adequately described in the material and methods section under the subheading (2.1 Study Design and Setting, lines 75-86 of the originally submitted manuscript). It was mentioned that: This is a retrospective cross-sectional study conducted at a tertiary university hospital. Pediatric patients ≤ 19 years with confirmed SARS-CoV-2 infection and hospitalized at KAUH between April 2020 and February 2022, were included in this study. COVID-19 positive patients with incomplete data in their medical records were excluded. The admission policy for pediatric COVID-19 required confirmed SARS-CoV-2 infection in symptomatic cases with clinical suspicion of COVID-19 or in asymptomatic cases with history of contact with SARS-CoV-2 positive cases and underlying comorbidity considering that a high-risk factor for severe COVID-19 disease.  The place of admission either to the pediatric ward or the pediatric intensive care unit (PICU) was determined by the attending physicians according to the severity of the case.

  1. In table 1, main presenting complaints: Fever, Fever and gastrointestinal (GIT) symptoms and Gastrointestinal symptoms. These symptoms somehow overlap but I cannot track how.

Response/Reply:

Authors were very keen to describe accurately and in details the symptoms of patients on initial presentation or the main presenting complaint and this shows how far the initial presentation or symptoms of COVID-19 infection are variable between patients as some may present with only fever, some with both fever and GIT symptoms at the same time and some with only GIT symptoms at the first presentation or the beginning of their COVID-19 illness. So, the categorization of initial symptoms of COVID-19 on the first presentation or according to the main complaint for example into fever alone, fever with GIT symptoms and GIT symptoms alone, was done not make a confusion or overlap problem but to show that COVID-19 may even present initially with fever alone, GIT symptoms alone or with both fever and GIT symptoms. This is important to alert physicians or clinician to the variable initial presentations of COVID-19 or in other words, COVID-19 may be suspected even in the absence of the usual known symptoms of fever and respiratory symptoms and may present initially or at the beginning with GIT symptoms as a main complaint for example.   

  1. How 10 children with gastrointestinal symptoms be only 0.6%?

Response/Reply:

Thanks for the reviewer for noticing this error. Ten children out of a total of 94 children equals 10.6% and certainly not 0.6%. This typographic error may be just happened during conversion of the original manuscript word file to the journal format file. So, authors confirm that the correct % is 10.6%.

  1. Figure 1: Pi chart suggest that neurological symptoms sum to 100% but they are not. Does neurological symptoms overlap with one another?

In figure 1, each neurological symptom was correlated or considered in relation to the total number of COVID-19 patients of 94 to show the percentage of each neurologic symptom as a percentage of the whole group of patients with COVID-19 infection. However, thanks to the reviewer for this observation and to avoid any confusion and to make everything clear, the sum of frequency distribution of neurological symptoms was readjusted to 100% and necessary corrections in relevant sites of the manuscript were undertaken.

  1. In table 2, why authors supply all the lab findings in the whole group when paper is about neurological manifestation. The authors do not further discuss this data at all.

Response/Reply:

The authors provided the lab findings of the whole group to present briefly the lab findings in pediatric COVID-19 patients in general whether with or without neurological manifestations because this is the major concern of the whole world now to know as much as possible data about COVID-19 infection including the lab data. Unfortunately, the reviewer didn’t notice that authors further discussed the important findings of leukocytosis and lymphopenia and the relation of lymphopenia to severe pediatric COVID-19 especially in patients with life threatening neurologic involvement as it was mentioned in the discussion section (lines: 263-267 of the originally submitted manuscript) that: Regarding investigations, 17% and 34% of the studied group had leukocytosis and lymphopenia respectively but with no statistically significant difference between COVID-19 patients with and without neurological manifestation. A large study performed at 61 hospitals which included 1695 patients demonstrated that lymphopenia is a common finding in COVID-19 pediatric patients especially those with life threatening neurologic involvement [21].  

  1. In table 3, second row: COVID-19 cases with neurologic manifestations - FEVER 6 children. If these are children with neurologic manifestation and do have fever why they are not in the fever and neurological symptoms group?

Response/Reply:

Authors mentioned the 6 children who had only fever at initial presentation (as written in table 3: Symptoms at initial presentation) which means that these children developed neurological manifestations later on during the disease course. Another 8 patients had both fever and neurological symptoms at initial or first presentation. As explained above in answer to query number 2: Authors were very keen to describe accurately and in details the symptoms of patients on initial presentation or the main presenting complaint and this shows how far the initial presentation or symptoms of COVID-19 infection are variable between patients with neurological manifestation as some may present with only fever, then they may develop neurological manifestations while some patients initially present with both fever and neurological manifestations at the same time. These show that neurological symptoms/manifestations can be present from the start or develop during COVID-19 disease course.

  1. According to table 1 there were only 8 children presented with neurological symptoms and finally 29. Did these symptoms occur later in the course of disease?  Again, I think the group definition is not clear.

Response/Reply:

The 8 children presented with neurological symptoms at the first or initial presentation as mentioned in table 1 as the (Main presenting complaint). Finally, 29 patients had neurologic symptoms because these symptoms occurred later in the course of disease as the reviewer has expected. For more clarity, the words (symptoms at initial presentation) were added beside main presenting complaint in table 1.

Reviewer 2 Report

The authors present a single center retrospective study on neurologic manifestations on hospitalized pediatric patients with COVID-19. The time frame was from April 2020 to February 2022. Several waves of COVID-19 occurred during this time period making antibody testing progressively unreliable to detect recent COVID-19 infection. The authors restricted their population to patients who tested positive by nasal RT-PCR. This limits detection of remote infection but may have limited detection of some MIS-C cases. This may explain why only one patient was found to have MIS-C. 

The advantage of this study is a report of neurologic disease in pediatric patients with COVID-19 in a different geographic population than previously reported. 

Patients included were only tested by RT-PCR. The authors should comment on why antibody testing was not done and how this may have affected their results, especially in detecting some cases of MIS-C. 

Author Response

Reply to Reviewer 2 Comments:

  1. The authors present a single center retrospective study on neurologic manifestations on hospitalized pediatric patients with COVID-19. The time frame was from April 2020 to February 2022. Several waves of COVID-19 occurred during this time period making antibody testing progressively unreliable to detect recent COVID-19 infection. The authors restricted their population to patients who tested positive by nasal RT-PCR. This limits detection of remote infection but may have limited detection of some MIS-C cases. This may explain why only one patient was found to have MIS-C. 

Response/Reply:

Thanks for the reviewer for this observation and as the reviewer mentioned: Several waves of COVID-19 occurred during this time period making antibody testing progressively unreliable to detect recent COVID-19 infection. So, this study included patients who tested positive by nasal RT-PCR which is the gold standard diagnostic approach for COVID-19 detection to have better confirmed diagnosis of recent SARS-Co-V2 infection with great precision according to the KAUH protocol. Although this may have limited detection of some MIS-C cases but the detection of MIS-C cases was not the target or main objective of the current study which focused mainly on the neurological manifestations in pediatric COVID-19 infection. Furthermore, apart from a possible limited COVID-19 infection diagnosis by serologic testing, the other required criteria or case definition for MIS-C which depend on the clinical data and laboratory derangements of multiorgan damage/failure were not met or evident except in only one patient of the studied group.

  1. The advantage of this study is a report of neurologic disease in paediatric patients with COVID-19 in a different geographic population than previously reported. 

Response/Reply:

Authors thanks and appreciate the recognition of this advantage by the reviewer. That it is why this advantage was added in the strength points of this study (Discussion section, lines 377-379 of the originally submitted manuscript).

  1. Patients included were only tested by RT-PCR. The authors should comment on why antibody testing was not done and how this may have affected their results, especially in detecting some cases of MIS-C. 

Response/Reply:

As mentioned in reply to query number 1, this study included patients who tested positive by nasal RT-PCR which is the gold standard diagnostic approach for COVID-19 detection to have better confirmed diagnosis of recent SARS-Co-V2 infection with great precision according to the KAUH protocol. Although this may have limited detection of some MIS-C cases but the detection of MIS-C cases was not the target or main objective of the current study which focused mainly on the neurological manifestations in pediatric COVID-19 infection. Furthermore, apart from a possible limited COVID-19 infection diagnosis by serologic testing, the required criteria or case definition for MIS-C which depend on the clinical data and laboratory derangements of multiorgan damage/failure were not met or evident except in only one patient of the studied group.

Round 2

Reviewer 1 Report

If authors believe that basic laboratory findings are important in this paper they should explain what they claim to be a normal value for WBC, neutrophils, lymphocytes ect as this value is very age dependent. Children in this study are aged 0 to 19, a wide range.

Author Response

Reply to Reviewer 1 Comments:

If authors believe that basic laboratory findings are important in this paper, they should explain what they claim to be a normal value for WBC, neutrophils, lymphocytes ect as this value is very age dependent. Children in this study are aged 0 to 19, a wide range.

Response/Reply:

The normal reference values/ranges for WBCs, neutrophils, lymphocytes etc… were considered in relation to age and sex (whenever applicable) from the following reference which was added in the footnote of table 2. This reference was added to the reference list and necessary amendments were done to the order of references in the text as well as in the reference list.

Lo, S.F.  Reference Intervals for Laboratory Tests and Procedures. In Nelson Text Book of Pediatrics, 20th ed.; Kliegman, R.M., Stanton, B.F., St. Geme, J.W., Schor, N.F., Behrman, R.E., Eds.; Elsevier, Philadelphia, USA, 2016; pp. 3464–3473.
